

# Evaluation of open and closed path sampling systems for determination
# of emission rates of $NH_3$ and $CH_4$ with inverse dispersion modelling
Yolanda Maria Lemes[a], Christoph Häni[b], Jesper Nørlem Kamp[a], Anders Feilberg[*a]
[a]Department of Engineering, Aarhus University, Gustav Wieds Vej 10D, 8000 Aarhus, Denmark.
[b]School of Agricultural, Forest and Food Sciences, Bern University of Applied Sciences, Länggasse 85, 3052
Zollikofen, Switzerland
*Corresponding author: email: af@bce.au.dk;   Telephone: +45 30896099
Declaration of interest: none
**Abstract**
The gas emission rates of ammonia ($NH_3$) and methane ($CH_4$) from an artificial source covering a surface
area of 254 $m^2$ were determined by inverse dispersion modelling (IDM) from point and line-integrated
concentration measurements with closed and open-path analyzers. Eight controlled release experiments
were conducted with different release rates ranging from $3.8 \pm 0.21$ to $17.4 \pm 0.4$ mg $s^{-1}$ and from $30.7 \pm$
1.4 to $142.8 \pm 2.9$ mg $s^{-1}$ for $NH_3$ and $CH_4$, respectively. The distance between the source and
concentration measurement positions ranged from 15 m to 60 m. Our study consisted of more than 200
fluxes averaged intervals of 10 min or 15 min. The different releases cover a range of different climate
conditions: cold (< 5°C), temperate (< 13 °C) and warm (< 18 °C). As the average of all releases with all
instrument types, the $CH_4$ recovery rate $Q_{bLS}/Q$ was $0.95 \pm 0.08$ (n = 19). There was much more variation
in the recovery of $NH_3$, with an average of $0.66 \pm 0.15$ (n = 10) for all the releases with the line-integrated
system. However, with an improved sampling line placed close to the source an average recovery rate of
$0.82 \pm 0.05$ (n = 3) was obtained for $NH_3$. Under comparable conditions, the recovery rate obtained with
an open-path analyzer was $0.91 \pm 0.07$ (n = 3). The effects of measurement distance, physical properties
of the sampling line, and deposition are discussed.
**Keywords:** Method validation, Ammonia, Methane, Inverse Dispersion Method, Backward Lagrangian
Stochastic, bLS



## 1 Introduction

The global agricultural system is currently facing one of its biggest humanitarian challenges: feeding the world's rising population while preserving the environment and climate for future generations (FAO, 2017). The agricultural sector is a major contributor to global greenhouse gas (GHG) emissions (15%) and ammonia ($NH_3$) emissions (64%) (OECD and FAO, 2019), leading to air pollution, climate change, deforestation, and loss of biodiversity (Aneja et al., 2009).

The European Union has established a reduction target for 2030 to reduce the GHG emissions by at least 55% (EEA, 2019), compared to 1990, and $NH_3$ emissions by 19% (NEC Directive 2016/2284), compared to 2005. Agriculture must contribute to GHG emission reductions, and valid estimates of GHG emissions are important for national inventories regulation strategies and for selecting efficient mitigation techniques.

Choosing the appropriate methodology to quantify gaseous emissions can be a challenge. In particular agricultural sources are challenging as the sources often are small and inhomogeneous, exhibit non-steady emissions over time (e.g. $NH_3$ emissions after slurry application (Hafner, 2018)) and are influenced by other sources in close vicinity. Most of the methodologies have restrictions on the measurement location and/or the source and involve complex instrumentation set-up (e.g., fast-response analyzers, measurements at multiple heights). The micrometeorological mass balance (MMB) method (Desjardins et al., 2004) requires measuring concentration at multiple positions several meters above the ground, which is a challenge for obtaining high time resolution and it ignores the horizontal turbulent transport (Hu et al., 2014). The tracer flux ratio method (TRM), which has also been used to measure agricultural emissions (Vechi et al., 2022; Fredenslund et al., 2019; Delre et al., 2018), is a relatively labor and cost intensive method typically with short intense measurement periods. In case of dynamic emissions, this is not sufficient for resolving the temporal variations in emissions over days or weeks.



The inverse dispersion method (IDM) based on backward Lagrangian Stochastic (bLS) dispersion
modelling (e.g. Flesch et al., 2004, 1995) has been widely used for the assessment of $NH_3$ and methane
($CH_4$) emissions from many agricultural sources: dairy housing (Bühler et al., 2021; VanderZaag et al.,
2014; Harper et al., 2009), cattle feedlot (McGinn et al., 2019; Todd et al., 2011; van Haarlem et al.,
2008; Flesch et al., 2007; McGinn et al., 2007), application of liquid animal manure (Kamp et al., 2021;
Carozzi et al., 2013; Sintermann et al., 2011; Sanz et al., 2010), grazed pasture (McGinn et al., 2011;
Voglmeier et al., 2018), rice field (Yang et al., 2019), lagoon (Ro et al., 2014; Wilson et al., 2001),
composting stockpiles (Sommer et al., 2004), agricultural biodigester (Baldé et al., 2016b; Flesch et al.,
2011), farm (Flesch et al., 2005) and stored liquid manure (Lemes et al., 2022; Baldé et al., 2016a; Grant
et al., 2015; McGinn et al., 2008).
IDM has been tested in controlled release experiments with different conditions: ground level
source without obstacles (Flesch et al., 2014; McBain and Desjardins, 2005a; Flesch et al., 2004), ground
level source surrounded by a fence (Flesch et al., 2005; McBain and Desjardins, 2005a), elevated source
(Gao et al., 2008; McBain and Desjardins, 2005a), multiple emission sources (Hu et al., 2016; Ro et al.,
2011; Gao et al., 2008) and to quantify the effect of $NH_3$ deposition (Häni et al., 2018).
IDM is a function of the geometry and location of source and downwind concentration sensor
(including height for the sensor) and the turbulence characteristics in the surface layer. The statistical
properties of the flow in the atmospheric surface layer for the IDM are defined by the friction velocity
($u^*$), roughness length ($z_0$), the Obukhov length (L), and wind direction (Flesch et al., 2004). Emissions
are derived from concentration measurements up- and downwind of the source, which could be
determined with point or line-integrated measurements from closed- or open-path analyzers. IDM
assumes an ideal atmospheric surface layer, which means i) a horizontally homogeneous and flat surface,
ii) homogeneity and quasi-stationarity with respect to the turbulence characteristics and iii) spatially
uniform emissions from a confined source (Flesch et al., 2004). Therefore, there should not be any
obstacles (e.g., trees, buildings) in close vicinity of the source to fulfil the required IDM assumptions.



Additionally, IDM has the limitation that there should not be any other sources of the same gas species
that affects up- and downwind concentration differently. The IDM is simple, flexible (Harper et al.,
2011), robust even in no ideal conditions and has a reported accuracy of $100 \pm 10\%$ when it is properly
used (e.g., place of instruments, filtering criteria) (Harper et al., 2010). Moreover, IDM is a direct
measurement method that does not alter the physical properties of the source, and it is applicable for both
small and large emissions of any shape of sources (Flesch et al., 2004) as opposed to indirect enclosure
methods (e.g. chambers measurements).
Concentration measurements are mostly done with an open-path optical system (e.g. Baldé et al.,
2018; Bühler et al., 2021) because long path lengths (>50 m) enable a higher emission plume coverage
and avoids internal surfaces (e.g. tubes, pumps) where $NH_3$ can adsorb (Shah et al., 2006; Vaittinen et al.,
2014). However, open-path has a limitation on low concentration measurements (<10 ppb for $CH_4$ and
$NH_3$) (Bai et al., 2022) and requires complex calibrations to reduce the uncertainty of the measurements
(Häni et al., 2021; DeBruyn et al., 2020). In addition, it requires intensive labor to move and optically
align the instruments to different positions depending on the predominant wind direction. Commercially
available open-path analyzers exhibit limitations with respect to acceptable detection limits (Häni et al.,
2021). Closed-path analyzers have rarely been used together with the IDM (Ro et al., 2011) due to its
limitation caused by adsorption of $NH_3$ in the system. In addition, closed path analyzers have only been
used for point measurements, which challenges the ability to catch the emission plume and makes it
sensitive to wind direction accuracy.
Data filtering is needed to ensure accuracy of the IDM, which is related to the meteorological
conditions (e.g., wind speed, atmospheric stability) and wind direction. The quality criteria for filtering
are based on the atmospheric conditions in a measurement interval to ensure the assumptions of the model
is adequately meet, which also lower the uncertainty of the resulting data. Different criteria have been
used in previous studies: Flesch et al. (2005) recommend to remove data where $u^* < 0.15$ m s$^{-1}$, $|L| < 10$ m
and $z_0 > 1$ m, whereas McBain and Desjardins, 2005 recommend $u^* < 0.19$ m s$^{-1}$, $|L| \le 3$ m and $z_0 > 1$ m.





Flesch et al. (2014) suggest the filtering criteria for the night of $u^* < 0.05$ m s$^{-1}$ and the gradient between
measured and MO-calculated temperature ($|\Delta\Delta T|_{thres}$) = 0.05 K. Bühler et al. (2021) removed data
where $u^* < 0.05$ m s$^{-1}$, $|L| < 2$ m, $z_0 > 0.1$ m, standard deviation of the horizontal wind components (u, v)
divided by $u^*(\sigma_{u,v}/u^*) > 4.5$ and Kolmogorov constant (C0) > 10.
This study aimed to assess the applicability and performance of a closed-path analyzer used with a
sampling system that allows for line integrated concentration measurements used with the IDM for
determining emission rates of $CH_4$ and $NH_3$. This novel measuring system will allow for measuring
emissions from sources with low emission rates and will have good flexibility for moving it around the
source depending on the wind direction in order to increase the probability of catching the emission
plume. This novel method is assessed by eight controlled releases of $CH_4$ and $NH_3$ combined with up- and
downwind measurements in different positions using point and line-average concentration provided with
closed- and open-path analyzers. The use of $CH_4$ and $NH_3$ and open- and closed-path systems to measure
concentration will give us an opportunity to: i) test the system of the line-average concentration
measurement with a closed-path analyzer; and ii) evaluate potential loss of $NH_3$ downwind from the
source by deposition and/or gas-to-particle conversion, processes that will not occur for inert $CH_4$. This
controlled-release study is the first to compare the performances of open-path and line-integrated closed-
path systems for measuring emissions of $NH_3$ and $CH_4$.

**2    Material and methods**
2.1    Site descriptions
From November 2019 to March 2022, eight controlled release experiments were performed at
different grassland sites under varying conditions (see Table 1). Five releases (I-DK to IV-DK and VIII-
DK) took place at AU campus Viborg, Denmark on two different fields (56°29'34.5"N / 9°34'28.3"E and
56°29'36.4"N / 9°34'15.9"E). Three releases (V-CH to VII-CH) were performed at Bern University of



Applied Sciences, Switzerland (46°59'35.1"N / 7°27'43.1"E). At all sites, the terrain was horizontally flat,
and the height of the canopy varied between 15 and 25 cm for the different experiments. Obstacles upwind
of the artificial source were more than 100 m away in all experiments. There were no significant sources
near the experiment sites.
2.2    Instrumentation
In this study, different models of cavity ring-down spectroscopy (CRDS) from Picarro (Picarro Inc.,
Santa Clara, CA, USA) were used to measure up- and downwind $NH_3$ and $CH_4$ concentration (Table 1).
Model G2201-i and model G4301 measure $CH_4$ concentration, G2103 measures $NH_3$ concentration, and
G2509 measures $CH_4$ and $NH_3$ simultaneously. The CRDS is a closed-path analyzer with continuous
absorption that measure concentrations at approximately 0.5 Hz. The CRDS analyzer consists of a laser and
an optical cavity chamber with highly reflective mirrors, which gives an effective path length of several
kilometers. The light is absorbed in the cavity, and the decay of light intensity is called the ring-down time,
which is directly related to the concentration of the specific compound. It has been frequently used to study
agricultural emissions (e.g., Kamp et al., 2021; Pedersen et al., 2020; Kamp et al., 2019; Sintermann et al.,

138    2011).

In experiments V-CH to VII-CH, the downwind $CH_4$ concentration was measured with three
GasFinder3 analyzers (GF3, Boreal Laser Inc., Edmoton Canada) and the downwind $NH_3$ concentration
with three miniDOAS instruments (Sintermann et al., 2016). The GF3 analyzer is an open-path tunable
diode laser device that measures line-integrated $CH_4$ concentrations over path lengths of 5 to 500 m (i.e.
single path length between sensor and retroreflector) with a temporal resolution of 0.3 to 1 Hz. The
retroreflectors used in the experiments were equipped with seven corner cubes, suitable for path lengths
around 50 m. The GasFinder devices have been widely used to measure emissions from different type of
agricultural sources with the IDM (Bühler et al., 2021; McGinn et al., 2019; VanderZaag et al., 2014; Harper
et al., 2010; Flesch et al., 2007). The performance of the GF3 instruments is discussed in detail by Häni et
al. (2021).



The miniDOAS instrument is an open-path device that measures $NH_3$, NO and $SO_2$ in the UV region
between 190 and 230 nm based on the differential optical absorption spectroscopy (DOAS; Platt and Stutz,
2008) technique. It provides path-averaged concentrations for path lengths between 15 m and 50 m, with
around 10 to 20 scans per second averaged over 1 minute. Ammonia emissions from agricultural sources
(Kamp et al., 2021; Kupper et al., 2021; Voglmeier et al., 2018) and from an artificial source (Häni et al.,
2018) have been measured with miniDOAS analyzers. Further details on the instrument is given in
Sintermann et al. (2016).
2.3    Gas release from an artificial source
The artificial source area had a gas distributor unit at the center and eight 1/4" polytetrafluoroethylene
(PTFE) tubes leave the distributor to get a circular shape of the source area. Each tube contained three
critical orifices (100 µm diameter, stainless steel, LenoxLaser, USA) in series with 3 m distance between
them. In total, the 24 orifices covered a circular area of 254 $m^2$.
Gas was released from a gas cylinder and the flow was controlled with a mass flow controller (in
Denmark: Bronkhorst EL FLOW, Ruurlo, Netherlands; in Switzerland: red-y smart controller, Voegtlin
Instruments GmbH, Aesch, Switzerland). The source height, the content of the gas cylinders, and the release
rate for each experiment are given in Table 1.





**Table 1** Date, gas cylinders description, ammonia and methane release rate (RR), source and canopy height, downwind
distance from source to instruments, type of system attached to the cavity ring-down spectroscopy (CRDS), and
instrumentation of each controlled release experiment (CRE). G2103, G2202-i, G4301 and G2508 are different CRDS
models, GF correspond to GasFinder and MD to miniDOAS.

| CRE | Date | Gas cylinder | | | NH$_3$ RR | CH$_4$ RR | Source height | Canopy height | Distance from source edge | System with CRDS | Instruments |
|---|---|---|---|---|---|---|---|---|---|---|---|
| | | Content | [bar] | total | [mg s$^{-1}$] | [mg s$^{-1}$] | [cm] | [cm] | [m] | | |
| I-DK | 29-11-2019 11:50 – 12:50 | 5% NH$_3$ and 95% N$_2$ ± 2%* | 62 | 1 | 4.6 ± 0.3 | - | 0 | 20 | 50 | Point 40℃ | 2 G2103 |
| II-DK | 29-11-2019 14:00-14:30 | 99% CH$_4$ and 1% N$_2$ ± 2%* | 62 | 1 | - | 30.7 ± 1.4 | 0 | 20 | 50 | Point 40℃ | G2201-i and G4301 |
| III-DK | 12-10-2020 11:45-15:15 | 5% NH$_3$ and 95% CH$_4$ ± 2%* | 62 | 1 | 3.8 ± 0.21 | 68.7 ± 3.7 | 0 | 25 | 35 - 60 | 16m line 40℃ (Line 1) | G2103, G4301 and G2508 |
| IV-DK | 20-07-2021 10:30-16:00 | 10% NH$_3$ and 90% CH$_4$ ± 2%* | 62 | 2 | 17.4 ± 0.4 | 142.8 ± 2.9 | 50 | 18 | 15-30 | 12m line 40℃ (Line 2) | G2103, G4301 and G2508 |
| V-CH | 09-10-2021 10:00-12:10 | 10% NH$_3$ and 90% CH$_4$ ± 2%+ | 27 | 2 | 15.2 ± 0.3 | 128.9 ± 2.7 | 0 | 15 | 15 - 30 - 60 | 16m line 40℃ (Line 1) | G4301, G2508, 3 GF and 3 MD |
| VI-CH | 09-10-2021 14:20-16:50 | 10% NH$_3$ and 90% CH$_4$ ± 2%+ | 27 | 4 | 13.2 ± 0.3 | 111.8 ± 2.2 | 0 | 15 | 15 - 30 - 60 | 16m line 40℃ (Line 1) | G4301, G2508, 3 GF and 3 MD |
| VII-CH | 09-10-2021 17:20-17:50 11-10-2021 15:10-16:20 | 10% NH$_3$ and 90% CH$_4$ ± 2%+ | 27 | 4 | 13.2 ± 0.3 | 111.8 ± 2.2 | 50 | 15 | 15 - 30 - 60 | 16m line 40℃ (Line 1) | G4301, G2508, 3 GF and 3 MD |
| VIII-DK | 22-04-2022 12:30-15:00 | 10% NH$_3$ and 90% CH$_4$ ± 2%* | 62 | 2 | 14.5 ± 0.3 | 118.9 ± 2.8 | 50 | 7 | 15 | 12m line 40℃ (Line 2) and 12m line 80℃ with heated inlets (Line 3) | 3 G2508 |

*Air Liquide, Horsens, Denmark
+Carbagas, Bern, Switzerland




2.4   Set-up
In the upwind position of all the experiments and in the downwind position of the I-DK and II-DK
experiment, the CRDS measured the concentration from a single point 1.5 m above ground through a
polytetrafluoroethylene (PTFE) tube that was insulated and heated to approximately 40℃. In the rest of the
experiments, the CRDS measured downwind concentration from a sampling line system of PTFE tubes
insulated and heated (40℃ or 80℃). In the III-DK, V-DK, VI-CH, and VII-CH experiment, the sampling
line system consisted of a 16 m tube with nine inlets, 2 m between each inlet (Line 1). In the IV-DK and
VIII-DK experiment, the sampling lines were 12 m long with seven inlets, 2 m between each inlet (Line 2
and Line 3). The inlets are made of critical orifices (0.25 mm ID for I-DK to VII-CH and 0.5 mm ID for
VIII-DK polyetheretherketone (PEEK)) that guarantee uniform flow through each inlet (Line 1, Line 2 and
Line 3). In the VIII-DK experiment, the sampling line system including the inlets was heated to 80°C (Line

184   3).

Figure 1 shows the position of the source area relative to the sampling position and the arrow
indicates the wind direction during the experiments. The downwind concentration were measured in one,
two or three distance (Table 1). In the V-CH, VI-CH and VII-CH, downwind concentrations were measured
at the same time at 15 m, 30 m and 60 m distance from the edge of the source with multiple GF3 and
miniDOAS instruments; one CRDS instrument was placed 15 m downwind (Figure 1). The distance
between the reflector and the laser/detector of the GF3 and miniDOAS at the downwind position parallel
to the CRDS sampling line was also 16 m. For the other two downwind positions the path lengths were 15
m and 50 m, respectively. The height of the measurement paths of all the open-path instruments were
between 1.2 and 1.5 m. The background concentration of $NH_3$ was stable with no sources in close vicinity,
thus in the three experiments, the average concentration of each instrument 10 min before the release of
each experiment was used as the $NH_3$ upwind concentration for the miniDOAS and the CRDS instruments
. In the V-CH, VI-CH and VII-CH experiment the measured $NH_3$ background concentration was 2.7 and
4.1 mg m$^{-3}$, and 2.1 and 4.8 mg m$^{-3}$ for the miniDOAS and the CRDS, respectively.



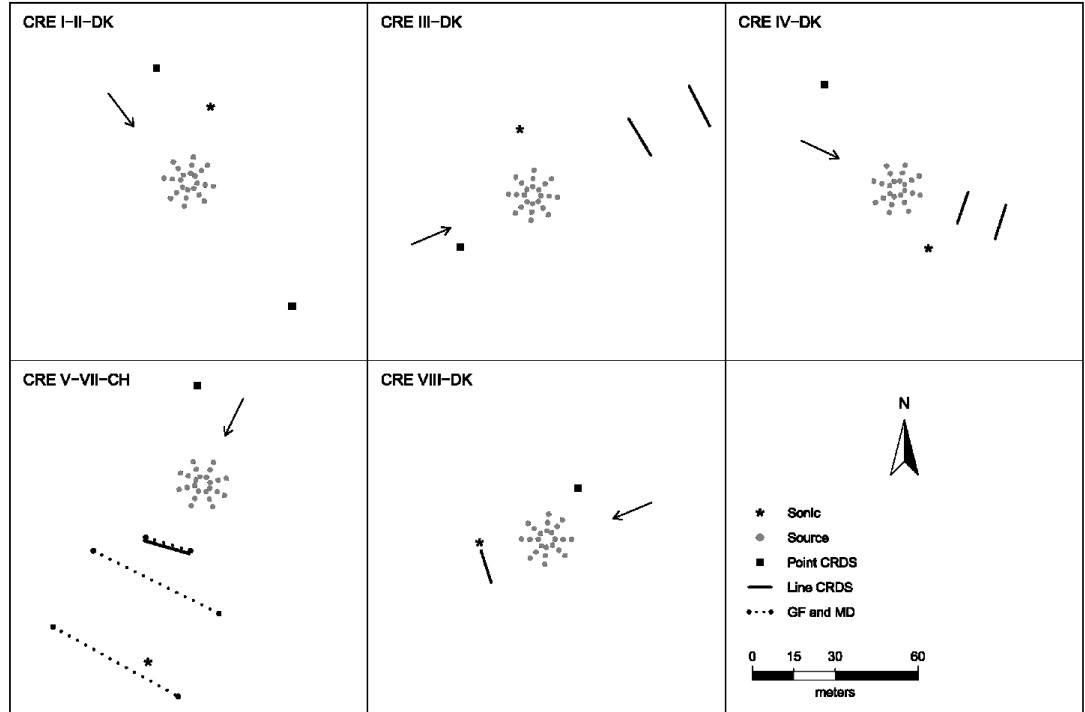


*Figure 1 – Position of the orifices of the artificial source, ultrasonic anemometer (sonic), and the*
*concentration analyzer used in the eight controlled release experiments (CRE) of this study. Three*
*types of analyzers have been used: cavity ring-down spectrometer (CRDS), GasFinder (GF) and*
*miniDOAS (MD). The arrow indicates the wind direction during each experiment.*
In Denmark, the three wind components were measured at 16 Hz with a 3D ultrasonic anemometer
(WindMaster, Gill, Hampshire, UK) at 1.5 and 1.7 m height. In addition to concentration and wind, air
temperature, and atmospheric pressure were also measured. In Switzerland, the wind components were
measured at 20 Hz with a 3D ultrasonic anemometer (WindMaster, Gill, Hampshire, UK) at 2 m height.
Air temperature and atmospheric pressure were obtained from a meteorological station nearby the
experiment site.
A Global Positioning System (in Denmark: GPS Trimbel R10, Sunnyvale, California, USA; in
Switzerland: GPS Trimble Pro 6, Sunnyvale, California, USA) was used to record the position of all
instruments and the individual critical orifices of the source.

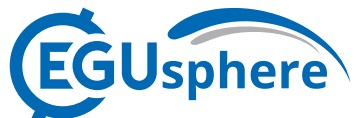

2.5  Inverse dispersion method
The measured gas emission rates (Q) from the artificial source were calculate in 15 min
(experiments conducted in Denmark) or 10 min average intervals (experiments conducted in Switzerland)
using the R  (R Core Team, 2018) package bLSmodelR (https://github.com/ChHaeni/bLSmodelR;
version 4.3) as described by Häni et al. (2018). The simulation was performed with six million backward
trajectories (N) and the source area defined as 24 individual circles of 5 cm radius as described by Häni et
al. (2018) with a high performance computer cluster (PRIME - Programming Rig for Modern
Engineering, Aarhus University).
The emissions rate (Q) is proportional to the difference between measured concentration downwind
($C_{downwind}$) from the source and the measured background concentration ($C_{upwind}$), and the dispersion
factor (D):

$$Q = \frac{C_{downwind} - C_{upwind}}{D} \tag{1}$$

The dispersion factor (D) is calculated as:

$$D = \frac{1}{N} \sum_{TDinside} \left| \frac{2}{w_{TD}} \right| \tag{2}$$

where N is the number of backward trajectories from the downwind analyzer location. The
summation refers to the trajectories touching inside the source area (TDinside) taking the vertical
velocity($w_{TD}$) at touchdown into account. The calculation of D includes determination of wind profiles
and turbulence statistics that are based on the Monin-Obukhov Similarity Theory (MOST).
2.6  Surface deposition velocity
Ammonia is a relatively short lived gas in the atmosphere and can either be chemically converted, or
subjected to dry or wet deposition. The dry $NH_3$ deposition rate is usually expressed with a deposition
velocity ($\upsilon_d^*$). It is a complicated phenomenon that is controlled by both atmospheric and land surface





processes (e.g. wind speed, solar radiation, vegetation reactivity). In this study, we assume $\upsilon_d$ takes place
uni-directionally and it is calculated with the canopy resistances:

$$\upsilon_d = \frac{1}{R_a + R_b + R_c} \tag{3}$$

where $R_a$ is the aerodynamic resistance, $R_b$ is the quasi-laminar boundary resistance and $R_c$ is the bulk
canopy resistance. $R_a$ is a function of wind speed and friction velocity (Baldocchi et al., 1987) that is
included in the bLS model, therefore Eq. 3 can be simplified as:

$$\upsilon_d^* = \frac{1}{R_b + R_c} \tag{4}$$

According to Garland (1977), $R_b$ can be calculated with Eq. 5 as a function of the roughness length
($z_0$), the friction velocity ($u^*$), the kinematic viscosity of air ($\nu$) and the molecular diffusivity of $NH_3$ in
air ($\delta_{NH_3}$).

$$R_b = \frac{1.45 \left(\frac{z_0 u_*}{\nu}\right)^{0.24} \left(\frac{\nu}{\delta_{NH_3}}\right)^{0.8}}{u^*} \tag{5}$$

Regarding $R_c$, it is related to the chemical characteristics of the studied gas and the characteristics of
the leaf (e.g. type, size). There are different models to calculate $R_c$. Due to the complexity and the
uncertainty of the determination of the resistance, $R_c$ was calculated following the same procedure as by
Häni et al. (2018) with the bLSmodelR. It was assumed that $Q_{bLS}/Q < 1$ was solely due to dry deposition.
A similar approach is used here, where 12 values of $R_c$ from 0 to 500 s m$^{-1}$ were tested in the bLS model
that includes ammonia deposition to estimate the $R_c$ giving $Q_{bLS}/Q = 1$ in all intervals. This was done with
linear interpolation between the two points closest to $Q/Q = 1$. Using this estimated $R_c$ and the calculated
$R_b$ value for each interval, $\upsilon_d^*$ was estimated for all intervals with all instruments. The $\upsilon_d^*$ values are
compared to previously reported values for $NH_3$.





Another approach for calculating the $R_c$ is with empirical equation, which will be used for calculating
values for $\upsilon_d^*$. These calculated values will be compared to the values obtained with the bLS model. It is
assumed that $R_c$ unidirectional and equal to the sum of the stomatal resistance $R_s$ and the cuticular
resistance $R_w$, see Eq.6.

$$\frac{1}{R_c} = \frac{1}{R_s} + \frac{1}{R_w} \tag{6}$$

The stomatal resistance $R_s$ is calculated with equation Eq.7 (Wesely, 2007):

$$R_s = R_{s(min)} \left[ 1 + \left( \frac{200}{SR + 0.1} \right) \right]^2 \frac{400}{T_s(40 - T_s)} \tag{7}$$

where $R_{s(min)}$ is minimum bulk canopy $R_s$ for water vapour that is assumed to be equals to 250 s
m$^{-1}$ (Lynn and Carlson, 1990), SR is the solar radiation, and $T_s$ is the soil temperature.
The cuticular resistance is calculated with Eq. 8 (Massad et al., 2010):

$$R_w = \frac{R_{w(min)} \; e^{a \, (100-RH)} \; e^{0.15T}}{(LAI)^{0.5}} \tag{8}$$

where $R_{w(min)}$ is the minimum cuticular resistance, a is an empirical factor, RH is the relative
humidity, T is the air temperature, and LAI is the leaf are index. The parameters $R_{w(min)}$ (10 s m$^{-1}$), a
(0.110) and LAI (2 m$^2$ m$^{-2}$) were obtained from Massad et al., 2010, Table 1.

## 3    Results and discussion

### 3.1    Recovered fractions of Ammonia and Methane

The accuracy of the bLS model is evaluated by the recovered NH$_3$ and CH$_4$ fractions, Q$_{bLS}$/Q, and
the standard deviation $\sigma_{QbLS/Q}$ ($\pm$) for all the releases. In all experiments except I-DK and II-DK (Table 1),
NH$_3$ and CH$_4$ were released simultaneously. The use of these two gases give us the additional opportunity





to assess potential loss of $NH_3$ downwind from the source by deposition or gas-to-particle conversion,
processes that will not occur for $CH_4$ due to its inertness. As the average of all releases and measurement
systems, the $CH_4$ recovery rate was $0.95 \pm 0.08$ (n = 19) (Figure 4). This recovery is similar to $0.93 \pm 0.14$
(n = 8) observed by Gao et al. (2008) with a different controlled releases configuration and ground-level
sources. There was more variation in the recovery of $NH_3$, with an average of $0.66 \pm 0.15$ (n = 10) for all
the releases with the line-integrated system. However, the improved sampling lines (Line 2 and 3) placed
at 15 m downwind from the source had an average recovery of $0.82 \pm 0.05$ (n = 3) for $NH_3$ (Figure 2).
Under comparable conditions, the $NH_3$ recovery rate obtained with the miniDOAS (MD) was $0.91 \pm 0.07$
(n = 3). Häni et al. (2018) observed almost the same recovered fraction, $0.91 \pm 0.12$, at 15 m from the
edge of the source with the MD. The recovery rates of all experiments in this study are shown in Figure 2,
Figure 3 and Figure 4, whereas climate conditions such as wind direction, friction velocity $u_*$, air
temperature, relative humidity (RH), soil temperature and solar radiation (SR) from each experiment are
presented in Table 2. I-DK and II-DK were conducted during cold conditions ($\sim 5°C$) with RH ranging
from 65 % to 71 %, whereas IV-DK and VIII-DK were conducted in warm conditions ($14 – 18°C$) with
RH between 48 % and 63 %. The other releases were conducted under moderate temperature conditions
($10 – 13°C$) with RH between 39% and 89%.

Additional information on the atmospheric conditions, weather conditions, and recovery fraction

rates for each average time interval for each release are shown in Table S1 in the Supplementary
Information.



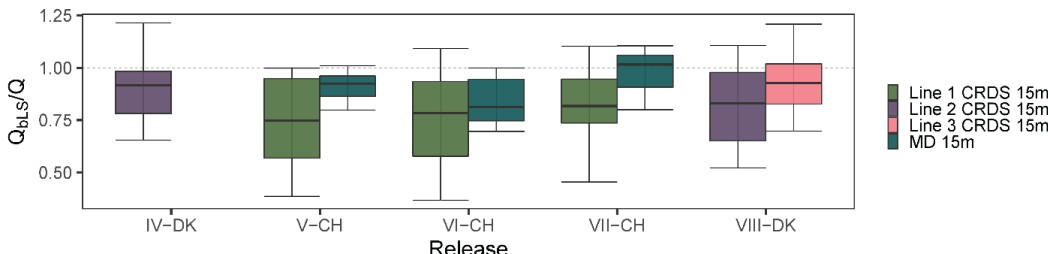

*Figure 2 .- The recovered fractions $Q_{bLS}/Q$ of ammonia from the releases where line 1, 2 and 3, and miniDOAS (MD) are placed 15 m from the edge of the source. Line 1 had a length of 16 m, and it was heated to 40 °C. Line 2 had the same temperature as Line 1, but it was 12 m long. Line 3 had the same length as Line 2, but was heated to 80 °C.*

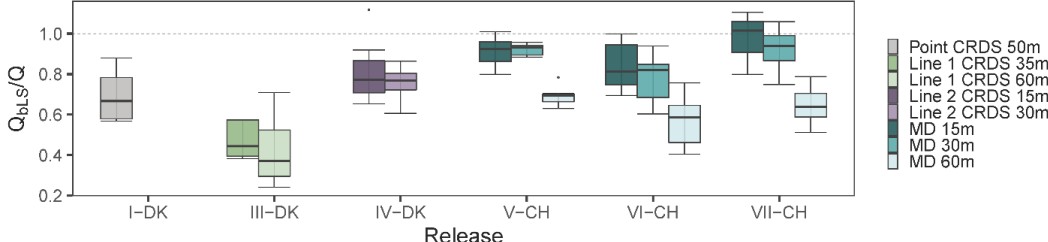

*Figure 3 .- The recovered fractions $Q_{bls}/Q$ of ammonia from the releases where point concentration measurement was used and where concentration downwind distance was measured in two or three distances from the edge of the source. Line 1, Line 2 and Line 3 are described in the Figure 2 caption.*

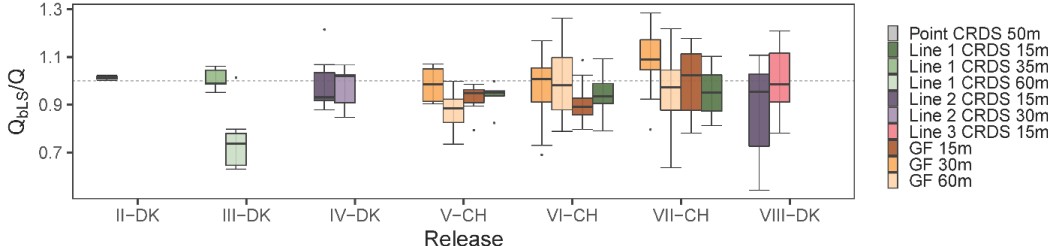

*Figure 4 .- The recovered fractions $Q_{bLS}/Q$ of methane for each release and analyzer. The downwind distance from the source to the analyzer is indicated in the legend. Line 1, Line 2 and Line 3 are described in the Figure 2 caption.*





Table 2 – Atmospheric and weather conditions in terms of friction velocity ($u^*$), wind speed (WS), air
temperature ($T_{air}$), air pressure ($P_{air}$), soil temperature ($T_{soil}$), solar radiation (SR) and relative humidity (RH)
during each release of this study.

| Release | $u^*$ [m s$^{-1}$] | WS [m s$^{-1}$] | $T_{air}$ [°C] | $P_{air}$ [hPa] | $T_{soil}$ [°C] | SR [W m$^{-2}$] | RH [%] |
|---|---|---|---|---|---|---|---|
| I-DK | 0.23 ± 0.05 | 2.4 ± 0.5 | 4.5 ± 0.3 | 993.6 ± 0.3 | 5.5 ± 0.1 | 167.5 ± 34.4 | 69.6 ± 1.7 |
| II-DK | 0.19 ± 0.03 | 1.8 ± 0.1 | 4.8 ± 0.1 | 995.4 ± 0.1 | 5.6 ± 0.1 | 117.1 ± 8.4 | 64.7 ± 0.2 |
| III-DK | 0.22 ± 0.03 | 2.2 ± 0.3 | 11.7 ± 3.1 | 1005.3 ± 0.3 | 10.3 ± 0.2 | 139.5 ± 50.7 | 76.7 ± 1.3 |
| IV-DK | 0.45 ± 0.04 | 5.1 ± 0.5 | 17.9 ± 0.4 | 1009.1 ± 0.1 | 17.6 ± 0.2 | 378.8 ± 152.1 | 66.7 ± 2.4 |
| V-CH | 0.36 ± 0.03 | 4.5 ± 0.1 | 9.4 ± 0.3 | 958.9 ± 0.1 | 11.8 ± 0.0 | 238.3 ± 47.8 | 86.5 ± 1.7 |
| VI-CH | 0.20 ± 0.04 | 2.3 ± 0.4 | 11.1 ± 0.3 | 959.6 ± 0.0 | 12.7 ± 0.2 | 178.7 ± 41.4 | 75.5 ± 1.8 |
| VII-CH | 0.26 ± 0.08 | 3.2 ± 1.0 | 12.8 ± 0.9 | 959.4 ± 0.1 | 11.5 ± 0.9 | 340.8 ± 161.7 | 52.1 ± 13.2 |
| VIII-DK | 0.43 ± 0.05 | 4.7 ± 0.3 | 13.9 ± 0.6 | 1008.9 ± 0.2 | 8.7 ± 0.3 | 691.7 ± 53.2 | 51.4 ± 2.4 |


3.2 Sampling systems for closed-path measurement

Three different CRDS sampling line systems have been used from III-DK to VIII-DK. The difference

between the lines was the length and the heating temperature. Line 1 had a length of 16 m, and it was heated
to 40 ºC. Line 2 had the same temperature as Line 1, but it was 12 m long. Line 3 had the same length as
Line 2, but was heated to 80 ºC, and the critical orifices have a higher inflow than Line 1 and Line 2 (see
section 2.4 Set-up). We expect that decreasing the length and increasing the heating temperature of the line
will improve $Q_{bLS}/Q$ for $NH_3$ (no expected effect for $CH_4$) by avoiding adsorption and reducing the response
time in the sampling line.

Line 1 was used with the source at ground level and elevated (Table 1), whereas the other two lines

only with the source elevated. When the source was at ground level, Line 1 had a recovery rate ranging
from 0.42 ± 0.17 to 0.60 ± 0.10 and from 0.75 ± 0.13 to 1.01 ± 0.05 for $NH_3$ and $CH_4$, respectively. The
lowest and the highest $NH_3$ recovery rate of Line 1 are directly related to the furthest (60 m) and the shortest
(15 m) downwind distance measurement from the source. In addition, the standard deviation $\sigma_{QbLS/Q}$ at the
furthest position is higher than at the closest position, which is in accordance with the results from Häni et
al. (2018). High uncertainty of the $Q_{bLS}/Q$ is related to a smaller difference in concentration between
downwind and background concentrations and due to smaller D-values (Häni et al., 2018). This is also the



reason for the low $CH_4$ recovery rate of Line 1 in III-DK at 60 m ($0.75 \pm 0.13$), downwind concentration is
only 4 – 10% higher than upwind concentrations since this is one of the lowest $CH_4$ releases rate (Table 1).
This is in line with Coates et al. (2021), who observed that the bLS model underestimated 49% of $CO_2$
released at 50 m fetch distance partially because the measured downwind concentration was close to the
background level. Therefore, in this study, the accuracy of $Q_{bLS}$ is mainly influenced by the uncertainty of
the concentration measurement, hence the downwind distance is limited by the properties of the gas
analyzers and the size of the emission strength of the source. This means the system can be limited in use
if the emission source has a large height and low emission strength where, as a rule of thumb, measurements
should be conducted at a distance from the source at least 10 times the height of the source (Harper et al.,

2011).

In VII-CH, Line 1 was used with the source elevated and had a recovery rate of $0.69 \pm 0.12$ for $NH_3$

and $0.95 \pm 0.10$ for $CH_4$. Line 2 had a numerically higher recovery rate than Line 1, ranging from $0.76 \pm$
$0.08$ to $0.81 \pm 0.16$ and from $0.89 \pm 0.20$ to $0.99 \pm 0.12$ for $NH_3$ and $CH_4$, respectively in IV-DK and VIII-
DK. As expected, the lowest $NH_3$ recovery rate of Line 2 was at the furthest downwind measurement
position (30 m). The length of the line appears to affect the $NH_3$ recovery rate; this might be due to the
increased surface area that $NH_3$ can adsorb to stick, and there is a lower flow in each of the critical orifices
that decreases the response time of the system (Shah et al., 2006; Vaittinen et al., 2014). Looking at the
measured $NH_3$ rates over time (Figure 5), higher emissions are reached with Line 3 for the first hour
indicating a faster time response compared to Line 2. However, after an hour there was not a clear difference
between the lines. The results indicate that increasing the sampling line temperature to 80 °C had a positive
effect on the recovery, which reached 87 % at a distance of 15 m. From the data obtained by the open-path
analyzer (MD), we can conclude that deposition can cause a reduction in recovery in the order of 2-16%
(Figure 2). Thus, the recovery obtained with the improved line (Line 3) approaches the recovery obtained
with the open-path analyzer. It should be noted that a direct comparison between Line 3 and the open-path
analyzer (MD) has not been made and further improvement can still be suggested for the CRDS sampling



line system. Specifically, increasing the flow on the sampling line will reduce $NH_3$ adsorption in the tubing
material. This can be achieved by increasing the flow through the tubing and the critical orifices but
maintaining an even flow distribution through the discrete sampling inlets in the sampling line must still be
maintained.

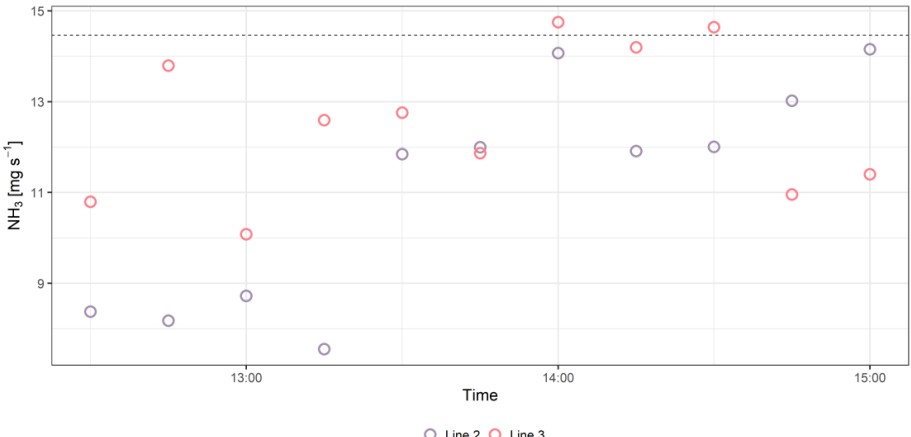


*Figure 5 .- Ammonia ($NH_3$) fluxes measured with Line 2 and Line 3 in 10 min intervals average in VIII-DK.*

The point CRDS system had a recovery rate of $0.70 \pm 0.22$ and $1.01 \pm 0.05$ for $NH_3$ and $CH_4$,
respectively. The benefit of the point CRDS system is mainly that increasing the flow in the tubing is less
limited, since there are no critical orifices for which equal flow must be maintained. However, comparing
point and line CRDS systems by the modelled concentration distribution (Figure 6), the line-integrated
measurement system covers a larger part of the emission plume from the source in a higher wind direction
range. In addition, a line-integrated measurement system can reduce uncertainty in the IDM (Flesch et al.,
2004), since it is less sensitive to error in the measured wind direction. This is in accordance with Ro et
al. (2011), who observed an almost double recovery value of a line-integrated measurement system for
$CH_4$ compared to a point measurement system using a photoacoustic gas monitor.



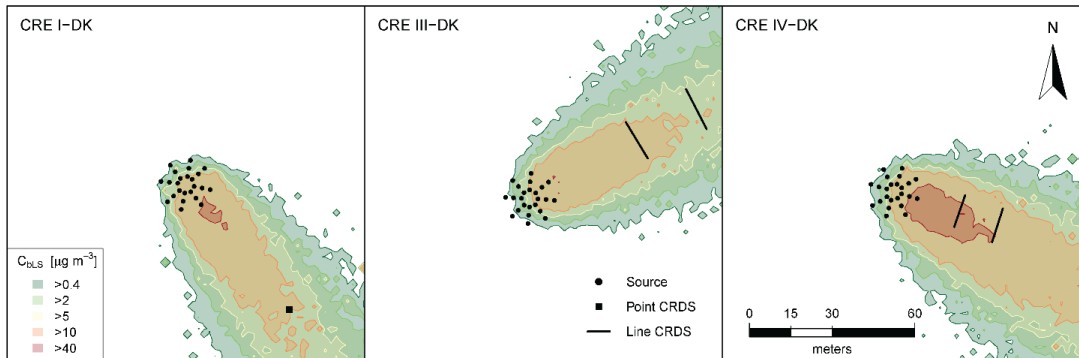


*Figure 6.- Contours of the modelled concentration distribution ($C_{bLS}$) for CRE I-DK, CRE III-DK and CRE IV-DK.*

3.3    Open-path measurement systems

The recovery rates for the GFs (CH$_4$) ranged from 0.87 ± 0.10 to 1.08 ± 0.15. In V-CH to VII-CH,

the corresponding standard deviation $\sigma_{QbLS/Q}$ of GF 15 m varies from 0.07 to 0.18, while Line 1 (placed
parallel to GF 15 m) ranges from 0.06 to 0.10. These standard deviations $\sigma_{QbLS/Q}$ are comparable with
those measured by Gao et al. (2009) (1.03 ± 0.16).

In V-CH and VI-CH (source at ground), the MDs (NH$_3$) had recovery rates ranging from 0.57 ±

0.12 to 0.93 ± 0.03. In VII-CH, MDs exhibit higher recoveries ranging from 0.64 ± 0.09 to 0.98 ± 0.10
since the source was elevated. Generally, it is recommended to do a release experiment above ground
level to reduce the probability of deposition close to the release area (McBain and Desjardins, 2005b). As
expected, the recovery rate decreased with downwind distance of the sampling position due to NH$_3$
deposition, which will be evaluated in section 3.4. Comparing MD at 15 m and Line 1 (placed in parallel)
in V-CH to VII-CH (Figure 2), the recovery rates are higher for MD. The highest difference between MD
and Line 1 was in V-CH, where there were the highest RH (87%). However, there are no clear patterns
explaining the difference between emissions from the different measurement systems based on
atmospheric conditions (Supplementary information, Figure S2). Although, the improved recovery with
Line 2 (0.81 ± 0.16) and Line 3 (0.87 ± 0.11) in IV-DK and VIII-DK could be influenced by the warmer



conditions and solar radiation (Table 2), it is plausible that the line improvements caused the increase. An
increased flow through the orifices and higher temperature of the sampling line will lead to less $NH_3$
adsorption thereby getting a better recovery from the release.
This results show the advantage of an open-path instrument compared to a closed-path instrument
to measure $NH_3$ emissions (Figure 2), since open-path avoids prolonged response caused by the
adsorption of $NH_3$ to sampling materials (Shah et al., 2006; Vaittinen et al., 2014). However, it is more
difficult to evaluate the quality of measurements by an open-path instrument due to complexity of the
calibration that depends on the path length between the sensor and reflector (DeBruyn et al., 2020) or the
need of another instrument for intercomparison (Häni et al., 2021). In addition, the closed-path system
presented in this study (line CRDS) is more flexible with respect to moving the sampling line around the
source depending on the predominant wind direction. This factor has different impact in different
countries, e.g. in the case in Denmark wind direction change quite more often than in Switzerland.
3.4     Surface deposition velocity
The corresponding surface deposition velocities ($\upsilon_d^*$) required to have a recovery rate $Q_{bLS}/Q = 1$
are presented in Figure 7. This approach assumes a complete recovery for each of the measurement
systems when taking deposition into account, which is not completely correct for closed-path sampling.
In the following, therefore, we refer to deposition velocity required to achieve $Q_{bLS}/Q = 1$ as the *apparent*
*deposition velocity* ($\upsilon_{ad}^*$). This is included to provide data on deposition velocities for ammonia for which
data is currently very limited. The recovery rates observed in Figure 3 show that the MD performed best,
whereas lower $Q_{bLS}/Q$ were seen in the sampling lines, thus the lowest $\upsilon_{ad}^*$ is expected from MD.
Additional information of $R_c$ and $\upsilon_{ad}^*$ for each time intervals in each experiment is shown in Table S1 in
the Supplementary Information. The apparent surface deposition velocities ranged from 0.2 to 2.2 cm s$^{-1}$
for open-path data and from 0.2 to 4.7 cm s$^{-1}$ for the line sampling, respectively. Häni et al. (2018)
reported $\upsilon_{ad}^*$ in the range from 0.3 to 1.1 cm s$^{-1}$. In all the releases where downwind concentrations were



measured at different positions, $\upsilon_{ad}^*$ appears to increase with distance increases, with many cases. For
example, in VI-CH, $\upsilon_{ad}^*$ is $0.7 \pm 0.4$, $0.8 \pm 0.4$ and $1.4 \pm 0.4$ cm s$^{-1}$ at 15 m, 30 m and 60 m, respectively.
This is in line with the outcome of Asman and van Jaarsveld. (1991); a significant fraction of the emitted
NH$_3$ is deposited near the source, which supports the regulations that do not allow livestock sources near
sensitive eutrophic ecosystems (NEC Directive 2016/2284).
In V-CH, VI-CH and VII-CH, $\upsilon_{ad}^*$ from Line 1 are 2.8, 2.2 and 5.1 times higher than MD at 15 m.
As expected $\upsilon_{ad}^*$ was higher for Line 1 as the Q/Q was lower for Line 1 compared to MD in these
experiments. Line 1 (VII-CH) had $\upsilon_{ad}^*$ of $1.6 \pm 0.9$, whereas Line 2 and Line 3 (VIII-DK) had $\upsilon_{ad}^*$ of 2.2
$\pm 1.4$ and $2.1 \pm 1.0$ cm s$^{-1}$, respectively, when measuring 15 m from the elevated source. During VII-CH
and VIII-DK the temperature differed 1°C and the relative humidity was approximately the same, but
wind speed and solar radiation differed (Table 2). However, comparing the apparent deposition velocities
from these experiments show comparable values for Lines 2 and 3, but higher values for Line 1. Overall,
the Q/Q values for Line 1 were worse than MD and Lines 2 and 3, which is reflected it in the higher
apparent deposition velocities.

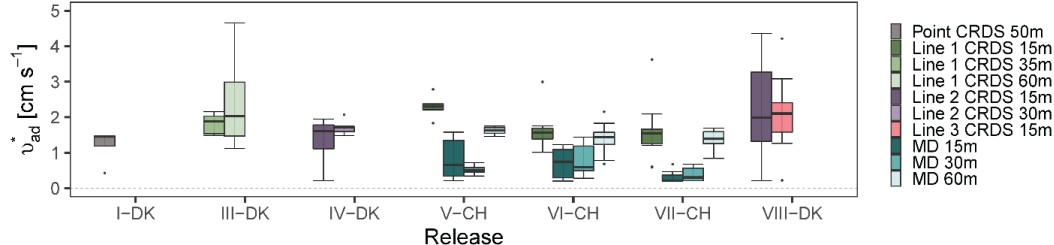


*Figure 7 .- Corresponding apparent surface deposition velocities ($\upsilon_{ad}^*$) required to have a recovery rate*
*$Q_{bLS}/Q$ closest to 1 in all the releases. All values are shown in Table S1 in the Supplementary Information.*
Many factors affect the deposition velocity, but it is possible to calculate $\upsilon_{ad}^*$ from empirical
models as explained previously (see section 2.6). *Figure 8* shows $\upsilon_{ad}^*$ for MDs in V-CH, VII-CH, and
VIII-CH compared to $\upsilon_{ad}^*$ calculated with the empirical models (equations 3-8). Using the empirical
models, $\upsilon_{ad}^*$ varies from 0.13 to 1.02 cm s$^{-1}$, increasing with the relative humidity (87%, 76% and 52%





RH in V-CH, VI-CH and VII-CH, respectively). The difference between the two ways of estimating $\upsilon^*_{ad}$
highlights the complexity and uncertainty for these methods. In addition, an artificial source has higher $\upsilon^*_d$
than what is expected from a real agricultural source (Häni et al., 2018). This is seen with the higher $\upsilon^*_{ad}$
values found in these experiments compared to the calculated values with the empirical models. The
height of the source might also have an influence on $\upsilon^*_{ad}$. This is indicated by the lowest $\upsilon^*_{ad}$ in VII-CH,
where the source was elevated compared to V-CH and VI-CH, where the gas was released on the surface.
Placing the source above ground level will reduce the obstacles (crop on the field) for gas dispersion,
reducing surface deposition. However, the bLS model does not consider the height of the source. For
example, evaluating emissions from the application of liquid animal manure (ground level source) or a
dairy housing (elevated source) will have different $\upsilon^*_{ad}$.

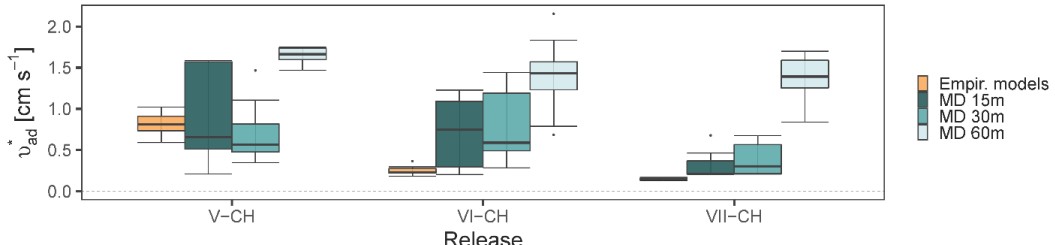


*Figure 8.- Corresponding apparent surface deposition velocities ($\upsilon^*_{ad}$) required to have a recovery*
*rate $Q_{bLS}/Q$ closest to 1 for miniDOAS (MD) in release V-CH, VI-CH and VII-CH and $\upsilon^*_{ad}$ calculated*
*with an empirical models.*

3.5    Sensitivity analysis
A sensitivity analysis of the bLS model was based on the resulting Q/Q ratio when changing the
inlet height of the analyzer and the wind direction offset compared to the valid measured values in release
VIII-DK. This was done for 11 fluxes average intervals of 15 min, where all emissions were estimated
again with the bLS model. For the assessment of the influence of the input for the measurement height all
other variables were kept constant. Likewise, for the influence of the wind direction, all other variables



448 were kept constant while the wind direction offset was changed. The results are presented in Figure S4 in

449 the Supplementary Information, where it can be seen that Q/Q was most sensitive to the changes in wind

450 direction offset, stressing the importance of the true offset in wind direction. Therefore, the wind direction

451 must be thoroughly evaluated for the accuracy of emission estimation since more or less trajectories have

452 touchdowns inside of the source area for the dispersion factor (Eq. 2). In addition, the uncertainty of Q/Q

453 ratio increases as wind direction offset increases. The emission estimation accuracy from point systems is

454 more sensitive to error in the measured wind direction (Flesch et al., 2004).

455 The accuracy of the emission estimation also depends on the detection limits of the concentration

456 sensor analyzer, especially when the downwind concentration is close to the background level, as it was

457 shown previously (see section 3.2). Therefore, it is recommended to conduct concentration and turbulence

458 measurements not far from the source but minimum 10 times the source height according to Harper et al.

459 (2011) at a known height to reduce the uncertainty of the calculated emissions rates.

460 **4 Conclusion**

461 Line-average concentration measurement with a closed-path analyzer is comparable with an open-

462 path system, as the average of all releases with all instrument types, the $CH_4$ recovery rate $Q_{bLS}/Q$ was

463 $0.95 \pm 0.08$ (n = 19). Under comparable conditions, an average $NH_3$ recovery rate of $0.82 \pm 0.05$ (n = 3)

464 and $0.91 \pm 0.07$ (n = 3) was obtained with the closed-path and open-path line integrated system,

465 respectively. The implementation of the new method presented in this study will enable measurement of

466 fluxes of multiple gases from different type of sources and evaluate the effects of mitigation strategies on

467 emissions. In addition, this method allows for continuous online measurements that resolve temporal

468 variation in $NH_3$ emissions and the peak emissions of $CH_4$.

469 A significant fraction of the emitted $NH_3$ is deposited near the source. Consequently, including the

470 deposition algorithm in the bLS model will have a greater influence on the emission evaluation at ground

471 level sources (e.g. application of liquid animal manure), compared to elevated sources (e.g. slurry tank).



The present study shows that the deposition algorithm included in the bLS model estimates correct NH$_3$
emissions that considers surface deposition. In addition, the wind direction must be thoroughly evaluated
for the accuracy of emission estimation with the bLS model.
**Acknowledgments**

This study was funded by the Ministry of the Environment and Food of Denmark as a service

agreement 2019-760-001136. Thanks to Simon Bowald for his great ideas and his help with designing
and building up the Line 3. Also thanks to technicians Martin Häberli-Wyss, Peter Storegård Nielsen,
Jens Kristian Kristensen, and Heidi Grønbæk for their invaluable help during the experimental part of the
study.

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
