# Peer review of "Evaluation of open and closed path sampling systems for determination"

_EGUsphere, 2022_

## Referee Comment (RC1)

This study presents results from a series of controlled release experiments, designed to evaluate the accuracy and precision of $NH_3$ and $CH_4$ flux estimates made using a stochastic Langrangian dispersion model and various measurement configurations. In particular, the use of open-path and closed-path measurement systems is compared for both gases. For reasons that are well described in the introduction to this study, such controlled release experiments can provide valuable insight to help guide both the measurement and modelling strategy adopted by future studies targeting "real world" emissions. The interpretation of the results presented here is complicated by the fact that each controlled release experiment did not involve the same set of measurement approaches, but I understand the logistical reasons for this. However, I feel that the presentation of both the results and conclusions needs some revision in order to help the reader to draw clear conclusions from this work. Overall, I suggest that the study is well suited for publication in AMT once the following points have been addressed.

The methods and instrumentation used in this study are generally clearly described. However, it would be good to see some more discussion of instrument calibration in this section. It is mentioned subsequently (L389) that ease of calibration is an important advantage of closed-path systems over open-path systems – I have no doubt that this is the case, but it is hard to assess this without more detail on the respective calibration strategies. From a modelling perspective, the bLS model including $NH_3$ deposition is crucial to the results presented in the paper, but there are no details of it given in the methods section. I appreciate that a full description is given in Häni et al. (2018), but I think it is important to include a basic summary here too (possibly including Eq. 17 and 18 from Häni et al. (2018)).

The presentation of results could in general be made clearer. As the authors state in the introduction, a key component of this study is the simultaneous release of $CH_4$ and $NH_3$, to disentangle methodological and depositional factors resulting in recovery rates less than 1. However, results from the two gases are not really considered together in section 3. For instance, it is concluded that heating Line 3 to a higher temperature resulted in reduced $NH_3$ loss in VIII-DK, but there is also an apparent improvement in $CH_4$ recovery rate using Line 3 as opposed to Line 2 for this experiment. How can this be explained? I suggest that the results section needs some reworking to take full advantage of the two-gas releases, so that the $NH_3$ results in each experiment are considered in the context of the corresponding $CH_4$ results.

It also took me a while to interpret Figs. 2-4. I would suggest combining the two $NH_3$ figures into a single figure (as has been done for $CH_4$). It may be even better to include the results for both gases in a single 2-panel figure, so that the results for each gas can easily be compared for the same experiment. Since the mean values are quoted in the text, it would be good to include these on the plots (as crosses perhaps). I would also state explicitly at the beginning of section 3.1 that the $Q_{bLS}$ values presented here for $NH_3$ do not take deposition into account.

The clarity of section 3.4 could also be improved. The statement that an artificial source has a higher deposition velocity than a real source needs more explanation and discussion. Is this the case in reality, or just a consequence of the way the bLS model is constructed? How does this impact the

interpretation of the results presented in this paper? It is unclear to me why the deposition velocity increases with distance from the source, or why this means that most $NH_3$ is deposited near the source (as stated in L408). More discussion is required to interpret the results shown in Figure 8 – what conclusions should we draw from the comparison against the empirical models?

The conclusion section does not currently summarise the key results from the paper particularly effectively. The opening statement is not supported by the average all-instrument $CH_4$ recovery rate that is quoted – this needs to be separated into the two instrument types (as it is for $NH_3$). I do not understand the statement "The present study shows that the deposition algorithm included in the bLS model estimates correct $NH_3$ emissions that considers surface deposition". My understanding of the results presented here is that the deposition velocity is estimated *by* the bLS model. This deposition velocity appears to vary with distance and differs from the empirical model results. So I'm not sure how it can be determined that this estimate is "correct"? It seems that the important conclusion of this comparison relies on an interpretation of the differences shown in Figure 8. I think there are important lessons to be drawn from this study regarding both measurement strategy and the modelling of $NH_3$ deposition velocity, but without a better synthesis of results it is currently hard for the reader to determine what these lessons are.

Specific points:

There are quite a few cases of incorrect number agreement (e.g. "the downwind concentration were") – I haven't listed them below, but it would be good if these could be corrected on the next proofread.

L16 – averaged *over* intervals

L77 – *non* ideal conditions

L196 – I'm confused by the fact that there are three experiments and two instruments listed here, but only four background values quoted. Should there be two more? It should be made clearer which value corresponds to which instrument-experiment combination, either by rephrasing the sentence or adding a table.

L249 – with *an* empirical equation

L251 – $R_c$ *is* unidirectional

L334 – If I'm interpreting the results correctly, there is no significant difference between the Line 2 $NH_3$ results at different distances. In which case I would suggest removing this sentence.

L336 – remove "stick"

L339 – it would be good to expand on why there was no difference after an hour

L347 – sentence needs rephrasing

L393 – sentence needs rephrasing

L406 – remove "increases, with many cases"

L416 – this discussion loses me; in what way are the values from Line 1 higher than those from Lines 2 and 3? This seems to directly contradict the values stated above.

L571 – The doi for Häni et al. (2018) is for a preprint – please replace with the doi for the final published article.

---

## Author Comment (AC1)

**Review of egusphere-2022-867**
Joseph Referee #1

This study presents results from a series of controlled release experiments, designed to evaluate the accuracy and precision of NH3 and CH4 flux estimates made using a stochastic Langrangian dispersion model and various measurement configurations. In particular, the use of open-path and closed-path measurement systems is compared for both gases. For reasons that are well described in the introduction to this study, such controlled release experiments can provide valuable insight to help guide both the measurement and modelling strategy adopted by future studies targeting "real world" emissions. The interpretation of the results presented here is complicated by the fact that each controlled release experiment did not involve the same set of measurement approaches, but I understand the logistical reasons for this. However, I feel that the presentation of both the results and conclusions needs some revision in order to help the reader to draw clear conclusions from this work. Overall, I suggest that the study is well suited for publication in AMT once the following points have been addressed.

The methods and instrumentation used in this study are generally clearly described. However, it would be good to see some more discussion of instrument calibration in this section. It is mentioned subsequently (L389) that ease of calibration is an important advantage of closed-path systems over open-path systems – I have no doubt that this is the case, but it is hard to assess this without more detail on the respective calibration strategies.

Description of the calibration carried out to the CRDS, GasFinders and miniDOAS were added in section 2.2. In addition, Figure S5 and Figure S6 in the Supplementary information were added as examples of a calibration. The advantage of the calibration of closed-path systems over open-path has been more clearly explained at the end of section 3.3.

From a modelling perspective, the bLS model including NH**3** deposition is crucial to the results presented in the paper, but there are no details of it given in the methods section. I appreciate that a full description is given in Häni et al. (2018), but I think it is important to include a basic summary here too (possibly including Eq. 17 and 18 from Häni et al. (2018)).

Equation 5 and Equation 6 together with a more thorough explanation about $NH_3$ deposition were added in section 2.6.

The presentation of results could in general be made clearer. As the authors state in the introduction, a key component of this study is the simultaneous release of CH4 and NH3, to disentangle methodological and depositional factors resulting in recovery rates less than 1. However, results from the two gases are not really considered together in section 3. For instance, it is concluded that heating Line 3 to a higher temperature resulted in reduced NH3 loss in VIII-DK, but there is also an apparent improvement in CH4 recovery rate using Line 3 as opposed to Line 2 for this experiment. How can this be explained? I suggest that the results section needs some reworking to take full advantage of the two-gas releases, so that the NH3 results in each experiment are considered in the context of the corresponding CH4 results. It also took me a while to interpret Figs. 2-4. I would suggest combining the two NH3 figures into a single figure (as has been done for CH4). It may be even better to include the results for both gases in a single 2-panel figure, so that the results for each gas can easily be compared for the same experiment. Since the mean values are quoted in the text, it would be good to include these on the plots (as crosses perhaps).

Figure 2, 3 and 4 were combined to one figure, which is now Figure 2. In addition, a new table (Table 2) was added with the information of $Q_{bLS}/Q_{NH3}$, $Q_{bLS}/Q_{CH4}$, and $Q_{NH3}/Q_{CH4}$. New $NH_3$ deposition velocities were calculated with an approach that assumes a recovery equals to the measured $Q_{CH4}$ for each of the measurement systems. This approach allows to see the improvement between Line 1, Line 2, and Line 3.

I would also state explicitly at the beginning of section 3.1 that the QbLS values presented here for NH3 do not take deposition into account.

Sentence added for clarification at the beginning of section 3.1.

The clarity of section 3.4 could also be improved. The statement that an artificial source has a higher deposition velocity than a real source needs more explanation and discussion. Is this the case in reality, or just a consequence of the way the bLS model is constructed? How does this impact the interpretation of the results presented in this paper? It is unclear to me why the deposition velocity increases with distance from the source, or why this means that most $NH_3$ is deposited near the source (as stated in L408). More discussion is required to interpret the results shown in Figure 8 – what conclusions should we draw from the comparison against the empirical models?

An explanation to the statement that an artificial source has higher $\upsilon_d^*$ than what is expected from some type of real agricultural source has been added in section 3.4. Explanation of the difference between the two ways of estimating $\upsilon_{ad}^*$ was also added in section 3.4.

We agree that it is unexpected that the deposition velocities increases with distance. The reason for this is presently unclear and should be investigated further. It is mentioned in section 3.4.

The conclusion section does not currently summarise the key results from the paper particularly effectively. The opening statement is not supported by the average all-instrument $CH_4$ recovery rate that is quoted – this needs to be separated into the two instrument types (as it is for $NH_3$).

The $CH_4$ recovery was also separated into the two instruments in the conclusion. In addition, an improvement of the conclusion was done.

I do not understand the statement "The present study shows that the deposition algorithm included in the bLS model estimates correct NH3 emissions that considers surface deposition". My understanding of the results presented here is that the deposition velocity is estimated *by* the bLS model. This deposition velocity appears to vary with distance and differs from the empirical model results. So I'm not sure how it can be determined that this estimate is "correct"? It seems that the important conclusion of this comparison relies on an interpretation of the differences shown in Figure 8. I think there are important lessons to be drawn from this study regarding both measurement strategy and the modelling of NH3 deposition velocity, but without a better synthesis of results it is currently hard for the reader to determine what these lessons are.

We have changed the conclusion to address this comment. "A significant fraction of the emitted $NH_3$ is deposited near the source. Consequently, including the deposition algorithm in the bLS model will have less bias in the emission evaluation at ground level sources (e.g. application of liquid animal manure), compared to elevated sources (e.g. slurry tank). The present study shows that the estimated deposition velocities are in the same order of magnitude in all the releases with some variation across the different approaches (instrument, distance, method)."

Specific points:

There are quite a few cases of incorrect number agreement (e.g. "the downwind concentration were") – I haven't listed them below, but it would be good if these could be corrected on the next proofread.

L16 – averaged *over* intervals. Added as suggested.

L77 – *non* ideal conditions. Added as suggested.

L196 – I'm confused by the fact that there are three experiments and two instruments listed here, but only four background values quoted. Should there be two more? It should be made clearer which value corresponds to which instrument-experiment combination, either by rephrasing the sentence or adding a table. Sentence added to clarify the background used in each experiment done in Switzerland in section 2.4.

L249 – with *an* empirical equation Added as suggested.

L251 – Rc *is* unidirectional Added as suggested.

L334 – If I'm interpreting the results correctly, there is no significant difference between the Line 2 NH3 results at different distances. In which case I would suggest removing this sentence. Good point, deleted as suggested.

L336 – remove "stick" Deleted as suggested.

L339 – it would be good to expand on why there was no difference after an hour.

L347 – sentence needs rephrasing Changed as suggested.

L393 – sentence needs rephrasing Changed as suggested.

L406 – remove "increases, with many cases" Deleted as suggested.

L416 – this discussion loses me; in what way are the values from Line 1 higher than those from Lines 2 and 3? This seems to directly contradict the values stated above.
We believe that the text is correct. But, we have changed the text to make it more clear and avoid confusion.

L571 – The doi for Häni et al. (2018) is for a preprint – please replace with the doi for the final published article. Changed as suggested.

---

## Author Comment (AC2)

**Review of egusphere-2022-867**
Anonymous Referee #2

Lemes et al. evaluated the performance of the inverse dispersion modeling with controlled releases of $NH_3$ and CH4 based on both open and closed path atmospheric sampling. The work used one single model, the backward Lagrangian stochastic model, to perform the analysis. Since the deposition of $NH_3$ on the surface may be significant and that of $CH_4$ not, the simultaneous measurements could provide a means of evaluating the deposition rates of $NH_3$. To this end, this work can be potentially quite interesting to the community. On the other hand, the manuscript can be better structured, and several important aspects should be clarified before the manuscript can be accepted for publication.

General comments:

1. Here the recovery rates were used to calculate the deposition velocity of $NH_3$. Although the authors are aware that this is not completely correct, and call it "apparent" deposition velocity, the assumptions behind this calculation have not been fully discussed, e.g., what are the sampling biases, the inverse modeling biases, the measurement biases that are related to and not related to sampling line deposition?

   A new approach has been used to calculate the $NH_3$ deposition velocities. This new approach assumes a recovery equals to the measured $Q_{CH4}$ for each of the measurement systems, which allowed to evaluate better the deposition related to the sampling line in section 3.4. In addition, a more detailed explanation regarding the calculations about $NH_3$ deposition velocities was added in section 2.6.

   Section 3.5 title was changed, the new title is "Uncertainties and sensitivity analysis". An entire paragraph was added to discuss precision for $CH_4$ and $NH_3$ concentration measurements was with the different. In addition, the sampling line adsorption bias related to the line-integrated system under the best conditions was also included.

2. The different causes for the mismatches in the calculated deposition rates have been presented; however, not sufficient efforts have been attempted to disentangle them. For example, the deposition of $NH_3$ on the sampling line could be directly compared, evaluated, and corrected for. Why has this not been done?

   The following text was added in section 3.4 to explain the difference between the calculate deposition rates. "The difference between the two ways of estimating $\upsilon_{ad}^*$ is not surprising since: i) bLS-derived deposition may be influenced by methodological uncertainties and therefore deviate from true deposition, ii) calculated resistances are associated with uncertainties due to estimations of physical parameters."

   In addition, the sampling line adsorption bias related to the line-integrated system under the best conditions was also evaluated in section 3.5.

3. A thorough analysis (or some sort of analysis) of the uncertainties of the inverse dispersion modeling is lacking. Note that inverse dispersion modeling has already been applied and evaluated in many other studies, e.g., Weller et al., 2018, Caulton et al., 2018, Shah et al., 2020, Andersen et al., 2021, Morales et al., 2022. It is well known that the inverse dispersion estimate based on one single measurement path is very uncertainty, which must be at least acknowledged.

   The uncertainties of the bLS model was already in the first version of the manuscript in the introduction. "The IDM is simple, flexible (Harper et al., 2011), robust even in non ideal conditions

and has a reported accuracy of $100 \pm 10\%$ when it is properly used (e.g., place of instruments, filtering criteria) (Harper et al., 2010)." The different studies mentioned were read, but unfortunately directly relation to this study was found. Regarding one single measurement path, it is known that it is an advantage to have more paths with more instrument for longer time measurement campaigns. However, for a release experiment with maximum 4h of measurement with stable conditions, one path placed downwind to the source was to sufficient to catch the downwind plume.

Minor comments:

P48: labor intensive and costly Changed as suggested.

P97: are adequately met Changed as suggested.

L106-109: It is not really novel. It is a novel method because it is the first time that a closed-path has been used with a line-integrated measurement system.

P129: …analyzers from Picarro Added as suggested.

P133: measures Changed as suggested.

Table 1: What's the uncertainty of the content of NH3 of the gas cylinder? The 2% uncertainty for both NH3 and N2? Yes. Table 1 shows the content of the gas cylinder used in each controlled release experiment.

L176-178: It's confusing here. What's the difference of a single point vs. the rest of the experiments? As is written, they all use PTFE tubes, insulated, and heated, and 40°C. Is 80°C the only difference? The difference between the point and line-integrated system is the number of positions where the gas sample is taken from. The point system has only one inlet, while the line-integrated has several. The inlets of the line-integrated system are made of custom-built critical orifices (0.25 mm ID or 0.5 mm ID polyetheretherketone) to guarantee similar inflow (<10 % variation) in each inlet. Three different versions of the line-integrated system (line) were built and used during this research. Both systems consist of PTFE and PVDF (only line 3) tube that was insulated and heated up to 40°C (point, line 1), 60°C (line 2) or 80°C (line 3). The length of line 1 was 16m, while 12m for line 2 and 3. More details about the measurement systems were added in section 2.4.

L213: calculated Changed as suggested.

L258: leaf area index Changed as suggested.

L306-312: This paragraph belongs to the method section. It is also included in the method section. But it was repeated in section 3.2 to help the reader to understand the difference between the three lines.

L385: These results Changed as suggested.

L406: Any correlation analysis result here? The following sentence shows the results.